# Atmospheric Ammonia Affects Myofiber Development and Lipid Metabolism in Growing Pig Muscle

**DOI:** 10.3390/ani10010002

**Published:** 2019-12-18

**Authors:** Shanlong Tang, Jingjing Xie, Sheng Zhang, Weida Wu, Bao Yi, Hongfu Zhang

**Affiliations:** 1State Key Laboratory of Animal Nutrition, Institute of Animal Science, Chinese Academy of Agricultural Sciences, Beijing 10093, China; long18763897938@163.com (S.T.); xiejingjing@caas.cn (J.X.); harrypolowwd87@163.com (W.W.); yibo33@126.com (B.Y.); 2Institute of Biotechnology, Cornell University, Ithaca, NY 14853, USA; sz14@cornell.edu

**Keywords:** ammonia, lipid metabolism, myofiber, meat quality, pig

## Abstract

**Simple Summary:**

The negative impacts of aerial ammonia on meat quality have been proven, but the mechanism still unclear. This study showed that different concentrations of ammonia exposure changed the meat quality of pigs indicated by the alteration of muscle fiber types and increased fat content. The data from RNA-Seq demonstrated that 10 mg/m^3^ ammonia exposure altered genes related to myofiber development (*MyoD1*, *MyoG*), whereas 25 mg/m^3^ ammonia affected genes associated with fatty acid synthesis and β-oxidation (*SCD*, *FADS1*, *FASN*, *ACADL*), which may explain the changes of meat quality to some extent. The alteration of lipid metabolism after ammonia exposure was inseparable from the metabolism of ammonia and glutamine in muscle.

**Abstract:**

Ammonia, an aerial pollutant in animal facilities, affects animal health. Recent studies showed that aerial ammonia negatively impacts meat quality but the mechanism remains unknown. To understand how ammonia drives its adverse effects on pig meat quality, 18 crossbred gilts were exposed to 0, 10 or 25 mg/m^3^ ammonia for 25 days. Ammonia exposure increased fat content in the *Longissimus dorsi* muscle, and meat color got lighter after 25 mg/m^3^ ammonia exposure. Analysis of MyHC isoforms showed an increased *MyHC IIx* but decreased *MyHC I* after ammonia exposure. Besides, muscular glutamine decreased significantly as aerial ammonia increased. Although hyperammonemia was reported to upregulate MSTN and inhibit downstream mTOR pathway, no changes have been found in the mRNA expression level of *MSTN* and protein expression level of mTOR signal pathway after ammonia exposure. RNA-Seq showed that 10 mg/m^3^ ammonia exposure altered genes related to myofiber development (*MyoD1*, *MyoG*), whereas 25 mg/m^3^ ammonia affected genes associated with fatty acid synthesis and β-oxidation (*SCD*, *FADS1*, *FASN*, *ACADL*). Collectively, our findings showed aerial ammonia exposure appears to regulate myofiber development and lipid metabolism in the skeletal muscle, which results in the negative impacts on meat quality in pigs.

## 1. Introduction

Ammonia is water-soluble noxious gas and is one of the most prominent aerial pollutants in animal farming facilities. Aerial ammonia can irritate eyes and damage the epithelial barrier of the airway [1]. On animal farms, bacterial decomposition of feed residues and nitrogen-containing excretions produces a large amount of ammonia emission. In intensive large-scale livestock husbandry facilities, aerial ammonia levels vary widely from 1.4 to 60.9 mg/m^3^ [2]. The daily ammonia concentration in some livestock houses for animals can exceed 35 mg/m^3^ in winter because of poor ventilation [3]. High levels of aerial ammonia can jeopardize the health and welfare of the animals, as well as farmworkers.

Besides its corrosive effects on the contact surfaces, ammonia is cytotoxic. In the brain, ammonia overloads can damage the development of the central nervous system [4]. Ammonia impairs protein synthesis in myocytes, resulting in sarcopenia [5,6]. By interacting with the third component of complement, ammonia activates the alternative complement pathway to induce tubule-interstitial inflammation in the kidney [7]. In farm animals, a few recent studies noted that ammonia exposure can impact carcass and meat qualities of broiler chickens in a dose-dependent manner. When broiler chickens were exposed 35 mg/m^3^, aerial ammonia changed meat color [8], increased drip loss [8], and decreased tenderness of breast meat in broiler chickens [9]. At a level of 52.5 mg/m^3^ ammonia, abdominal fat increased but fat deposition decreased in the muscle [10,11]. Meat quality is a complex trait closely depending on the physiology of skeletal muscles. Until now, the molecular and biochemical mechanism of aerial ammonia on muscle physiology is not yet completely known.

Although it is cytotoxic, ammonia is also an amino acid metabolite. In humans, an adult is estimated to produce 1000 mmol ammonia daily [12]. Liver, as the major organ, can incorporate ammonia into urea or synthesize glutamine from glutamate to detoxify ammonia [13,14]. Extrahepatic tissues, such as skeletal muscle and brain, can also detoxify ammonia through the glutamine synthesis. When liver function is impaired, skeletal muscle serves as the major organ to detoxify ammonia. During hyperammonemia, ammonia which has not been converted into glutamine activates the NF-κB pathway to up-regulate myostatin (MSTN) expression in the myocyte [15]. Myostatin is the most important negative regulator of muscle cell growth and differentiation. It not only regulates the thick filament myosin expression [16] but also interacts with the mTOR signaling to inhibit protein synthesis [5]. Therefore, hyperammonemia reduces skeletal muscle mass and leads to abnormal muscle development in patients with cirrhosis [6]. The question is whether aerial exposure of ammonia can increase serum ammonia to up-regulate muscle MSTN and hereby affect muscle physiology.

Therefore, taking the pig as a test case, the current study was designed to understand the molecular and biochemical mechanisms of aerial ammonia on meat physiology by investigating the ammonia metabolism, assessing the MSTN pathway, and further identifying the critical and associated pathways with the high throughput method of RNA-Seq.

## 2. Materials and Methods

### 2.1. Management of Animals

All animal procedures were approved by the Experimental Animal Welfare and Ethical Committee of Institute of Animal Science of Chinese Academy of Agricultural Sciences (IAS2017-2). Eighteen 70-day-old Yorkshire × Landrace sows were purchased from a commercial pig farm (Beijing Breeding Pig Co., Ltd., Beijing, China), randomly allocated into three groups, and maintained in three controlled-environment chambers (pigs were fed in individually pens). All animals were allowed free access to water and an amount of commercial feed (Appendix A) equal to 4–5% of body weight per day. Pigs in each chamber were exposed to 0 ± 2.5, 10 ± 2.5, and 25 ± 2.5 mg/m^3^ ammonia, respectively. Ammonia was first mixed with air and sent into each chamber via the ventilation system. Over the course of the 25-day period, the concentration of aerial ammonia was monitored by a ToxiRAE Pro Ammonia (NH_3_) Detector (RAE Systems, San Jose, CA, USA). The body weight of each pig was recorded weekly. At the end of the experiment, blood samples were collected from the precaval vein after 12 h fasting, and pigs were exsanguinated for *Longissimus dorsi* muscle tissue sampling. Muscle tissues were either subjected to meat quality assessment or frozen in lipid nitrogen and stored at −80 °C for further analysis of gene expression, RNA-Seq, western blot and free amino acids. Initial and final body weights were recorded to calculate the average body weight gain.

### 2.2. Assessment of Meat Quality

Muscle pH was quantified at 45 min using a pH-STAR instrument (Rudolph, Pöttmes, Germany) and meat color was measured using an OPTO-STAR system (Rudolph Matthäus). In order to determine drip loss, *Longissimus dorsi* muscle was cut into a cube of 5 cm × 4 cm × 1 cm, hung in a refrigerator at 4 °C for 24 h. The weight difference before and after hang-in the refrigerator was calculated as the drip loss. The muscle freeze-dried by freeze dryer was pulverized and mixed, then approximate 1 g of the freeze-dried muscle were taken for the measurement of fat content by Soxhlet extraction method, and the results were shown in the form of fat weight per 100 g freeze-dried muscle.

### 2.3. Measurement of Serum Ammonia and Urea

The concentration of serum ammonia and urea were measured with test kits according to the manufacturer’s instructions (Nanjing JianCheng Bioengineering Institute, Nanjing, China). Protein in the 200 μL of serum was first precipitated with reagents provided in the kit (cat # A086). Serum supernatant was subjected to Berthelot reactions and ammonia content was measured spectrometrically at 630 nm. Serum urea was determined by a colorimetric method (cat # C013), whereas the principle of that under the heating and acidic conditions urea nitrogen and diacetyl monoxime can develop an indigo blue color. The absorbance value was measured at a single wavelength of 520 nm to quantify urea content in the serum.

### 2.4. Profile of Free Amino Acids in Serum

Serum-free amino acids were analyzed by high-performance liquid chromatography (HPLC) using a reverse-phase C18 column after derivatization with o-phthaldialdehyde reagent using the adjusted method described by Dai et al. [17]. Briefly, 50 μL of serum was acidified with 200 μL of 1.5 M HClO_4_, followed by addition of 100 μL of 2 M K_2_CO_3_. The neutralized extract was determined for pre-column derivatization with *o*-phthaldialdehyde on a Waters HPLC system (Model Alliance e2695 Separation Module, Waters, Milford, MA, USA). Fluorescence of amino acid–*o*-phthaldialdehyde derivatives was detected using a Waters 2475 Multi λ Fluorescence Detector.

### 2.5. Free Amino Acids in Muscle

Muscle tissues (about 80 mg) were disrupted and homogenized in 600 μL of 8% sulfosalicylic acid using the high-through tissuelyser (SCIENTZ-48, SCIENTZ, Ningbo, China). After centrifugation at 15,000 rpm for 20 min, the supernatant was filtered through a 0.45 um Millex-LG filter (Millipore, Billerica, MA, USA). Each filtrate was diluted five times with acetonitrile, and centrifuged at 15,000 rpm for 20 min to collect the supernatant. Free amino acids were determined with ultra-performance liquid chromatography coupled with hybrid triple quadrupole mass spectrometry (UPLC-MS). A 5 μL supernatant was injected into Thermo Dionex Ultimate 3000 (Thermo Fisher Scientific, Waltham, MA, USA) equipped with an Agilent Poroshell 120 HILIC-Z column (3.0 × 100 mm, 2.7 um, Agilent Technologies, Santa Clara, CA, USA) to separate amino acids. The mobile phases consisted of 20 mM ammonium formate in water (pH = 3) and acetonitrile, and gradient elution was performed at a total flow rate of 0.4 ml/min. Amino acids were determined by Waters Xevo TQ-S (Waters, Milford, MA, USA) equipped with an ESI source. The spray voltage, vaporizer temperature and gas flow were set at 2.91 kV, 500 °C and 550 L/h. Standard curves were generated for each amino acid by serial dilution of the amino acid standard purchased from Sigma-Aldrich (Merck KGaA, Darmstadt, Germany). Good linearities were confirmed for all amino acids.

### 2.6. Quantitative Real-Time PCR for Genes Expression

Expressions of *MSTN, Smad2, FoxO1, MyHC I, MyHC IIa, MyHC IIx, MyHC IIb*, etc. (Appendix A) were quantified with qRT-PCR. Total RNA was isolated from muscle tissues with TRIzol reagent (Invitrogen, Carlsbad, CA, USA) according to the manufacturer’s instruction. The concentration of each RNA sample was quantified using the NanoDrop 2000 (Nanodrop Technologies, Wilmington, DE, USA). Possible contaminations from genomic DNA were eliminated by incubating with gDNA Eraser provided by the PrimeScript^TM^ RT reagent kit (Takara, Shige, Japan) before reverse transcription. qRT-PCR used the general protocol suggested by the kit (SYBR^®^ Premix Ex Taq^TM^, Takara). Specificity of each primer set was checked by the single peak in the melting curves after 40 PCR amplification cycles. Relative expression of each primer between control group and treatment group was calculated by 2^-^^△△Ct^ method using *GAPDH* as the reference gene.

### 2.7. Western Blot for Protein Expression

Proteins in the mTOR signal pathway and their phosphorylation were quantified by western blotting (WB). Briefly, total proteins of each muscle sample were extracted using RIPA lysis buffer (Beyotime Biotechnology, Shanghai, China) containing protease inhibitor cocktail (Roche, Basel, Switzerland) and phosphatase inhibitors (Solarbio, Beijing, China), and quantified with the BCA protein assay kit (Thermo Fisher Scientific, Waltham, MA, USA). Approximate 35 μg of denatured proteins were loaded to SDS-PAGE and then transferred to PVDF membranes using the wet method. For blotting, membranes were blocked in TBST with 5% skimmed milk for 2 h at room temperature, and incubated with primary antibodies rabbit anti-mTOR (Cell Signaling Technology, Beverly, MA, USA), 4EBP1 (Cell Signaling Technology), p-4EBP1 (Thr70, Cell Signaling Technology), p-P70S6K (Thr389, Cell Signaling Technology), P70S6K (Cell Signaling Technology) at a dilution of 1:1000 overnight at 4 °C. After several rinses, membranes were incubated with the second antibody (1:2000) for 40 min at room temperature. Targeted proteins were visualized using SuperSignal^®^ West Femto Maximum Sensitivity Substrate (Thermo) and a gel imaging system (Tanon Science & Technology, Shanghai, China). To normalize the differences in loading, mouse anti-GAPDH mAb (ZSGB-BIO, Beijing, China) was used as the internal control. The density of each band was quantified using the Image J software (MathWorks, Natick, MA, USA) and the quantity of each protein was expressed as the fold change to GAPDH.

### 2.8. RNA-Seq

Total RNA extracted from muscle was checked for integrity using a Bioanalyzer 2100 system (Agilent Technologies). The concentration of each RNA sample was quantified using Qubit^®^ RNA Assay Kit in Qubit^®^ 2.0 Fluorometer (Life Technologies, Carlsbad, CA, USA). Poly-T oligo-attached magnetic beads were used to purify mRNAs, which were chopped into short fragments and used as templates for cDNA synthesis. Sequencing libraries were generated using NEBNext^®^ Ultra™ RNA Library Prep Kit for Illumina^®^ (New England Biolabs, Ipswich, MA USA). Library qualities were assessed on the Agilent Bioanalyzer 2100 system. Index-coded samples were clustered using the cBot Cluster Generation System with TruSeq PE Cluster Kit v3-cBot-HS (Illumina, San Diego, CA, USA). Library preparations were then sequenced on an Illumina Hiseq platform and 125 bp/150 bp paired-end reads were generated.

Raw reads were firstly processed through in-house perl scripts to filter out reads containing adapter, ploy-N or any low-quality reads and generated clean data (clean reads). Swine reference genome index was built with Bowtie v2.2.3 and paired-end clean reads were aligned to the reference genome using TopHat v2.0.12. HTSeq v0.6.1 was used to count the reads mapped to each gene and FPKM of each gene was then calculated based on the length of the gene and reads count mapped [18]. Differentially expressed genes (DEGs) in 10 or 25 mg/m^3^ group against the control were determined using the DESeq2 R package (1.18.1) with Benjamini and Hochberg’s adjusted Q-value < 0.1 and fold-change ≥1.5 or ≤0.67. All DEGs were subjected to GO analysis by g: Profiler (https://biit.cs.ut.ee/gprofiler/index.cgi) to predict the function and metabolic pathways. DAVID (https://david.ncifcrf.gov/summary.jsp) was used to identify potential KEGG pathways. *p*-value < 0.1 was considered as the threshold of significance.

To evaluate the reliability of results from RNA-Seq, qRT-PCR was performed on 9 genes (4 genes for 10 mg/m^3^ and 5 genes for 25 mg/m^3^) with the Bio-Rad CFX96 Real-Time PCR Detection System (Applied Biosystems, Framingham, MA, USA) as described above.

### 2.9. Statistical Analysis

Data obtained from this research were analyzed by one-way ANOVA using SAS 9.2 software (SAS Institute, Cary, NC, USA) and body weight gain were analyzed by covariance analysis of GLM, then Duncan’s multiple comparison was used to compare results between every two groups. Differences between significant means were considered as statistically different at *p* < 0.05.

## 3. Results

### 3.1. Growth Performance and Meat Quality

In the test group of pigs exposed to 10 or 25 mg/m^3^ ammonia for 25 days, the body weight, average body weight gain, drip loss, or pH_45 min_ of *Longissimus dorsi* muscle showed no difference compared with those of control pigs (Table 1). However, meat color was found to be lighter in the pigs exposed to 25 mg/m^3^ ammonia (*p* = 0.001) than those exposed to 0 or 10 mg/m^3^ ammonia. Fat content was increased in the *Longissimus dorsi* muscle of pigs exposed to 10 and 25 mg/m^3^ ammonia (*p* = 0.007) as shown in Table 1.

Since meat quality traits were greatly influenced by muscle fiber compositions, we proceeded to analyze the four major sarcomeric myosin heavy chains (MyHC, Figure 1). 

In the pigs exposed to 10 mg/m^3^ ammonia, the proportion of *MyHC I* (*p* = 0.029) was reduced. In the pigs exposed to 25 mg/m^3^ ammonia, the proportion of *MyHC I* was also decreased (*p* = 0.029) but *MyHC IIx* fiber (*p* = 0.057) was increased. Interestingly, oxidative fibers (*MyHC I* and *IIa*, *p* = 0.013) were considerably decreased while its glycolytic isoforms (*MyHC IIb* and *IIx*, *p* = 0.013) were increased in pigs exposed to 10 and 25 mg/m^3^ ammonia.

### 3.2. Serum Ammonia, Urea and Free Amino Acids

In the pigs exposed to 10 and 25 mg/m^3^ ammonia, no changes were found in serum ammonia, urea and serum glutamine, glutamic acid as well as alanine (Table 2). Serum taurine was increased in the pigs exposed to 10 mg/m^3^ ammonia (*p* = 0.045) while tyrosine (*p* = 0.013) was also increased in the pigs exposed to 25 mg/m^3^ ammonia than that in control pigs.

### 3.3. Free Amino Acids in Muscle

As shown in Table 3, glutamine was decreased in the *Longissimus dorsi* muscle of pigs exposed to 10 and 25 mg/m^3^ ammonia (*p* = 0.008) than that in control pigs, whereas the level of glutamic acid was similar in all three groups. Isoleucine (*p* = 0.019) in muscle tissue gradually decreased with increasing concentration of ammonia exposure.

### 3.4. Expression of MSTN and Key Molecules Related to mTOR Pathway

To test whether *MSTN* is upregulated by aerial ammonia exposure, the level of *MSTN, Smad2* and *FoxO1* gene expression in *Longissimus dorsi* muscle was determined, but we found no difference among all treatment groups (Figure 2). 

Muscle expressions of mTOR, P70S6K, 4EBP1, p-4EBP1 and p-70S6K appeared not affected by ammonia exposure either (Figure 2).

### 3.5. Differentially Expressed Genes (DEGs) Associated with Ammonia Exposure

Because no changes in *MSTN* and its downstream pathways were observed in this study, RNA-Seq was then employed to screen for important genes associated with ammonia exposure. After quality control, each sample obtained 8.65~11.00 Gb clean reads. Over 76% clean reads were mapped to the swine reference genome. Among total mapping reads, the vast majority (79.50–83.90%) fell into annotated exons, 10.70–13.70% was within the large intergenic territory, and only 4.907.80% was located in introns. About 72.02% mapping reads were aligned to a unique gene and about 5.6% reads were mapped to multiple genes. Based upon FPKM of mapped genes, DEGs were analyzed using the DESeq2 method. A total of 123 DEGs (Q-value < 0.1, fold-change ≥1.5 or ≤0.67) have been found in the *Longissimus dorsi* muscle of pigs exposed to 10 mg/m^3^ ammonia, which included 85 down-regulated and 38 up-regulated genes (Figure 3A,C). In pigs exposed to 25 mg/m^3^ ammonia, 125 DEGs were found (Q-value < 0.1, fold-change ≥1.5 or ≤0.67), 42 down-upregulated genes and 83 up-regulated genes (Figure 3B,D). As shown in the Venn Diagram, DEGs in the pigs exposed to 10 mg/m^3^ ammonia had 5.1% overlaps with those identified in the pigs exposed to 25 mg/m^3^ ammonia comparing with control pigs (Figure 3E).

### 3.6. GO Annotations and KEGG Pathway Analysis

DEGs were subjected to GO annotations and KEGG pathway analyses. The DEGs identified in the pigs exposed to 10 mg/m^3^ ammonia were enriched in DNA binding function (GO: 007088, E-box binding; GO: 0008134, transcription factor binding; GO: 0003700, transcription factor activity, sequence-specific DNA binding) and acetylcholine-activated ion channel activity (GO: 0004889), involved in many biological processes related to development of muscle fibers (such as GO: 0048743, positive regulation of skeletal muscle fiber development; GO: 0010831, positive regulation of myotube differentiation; GO: 0045663, positive regulation of myoblast differentiation; GO: 0007517, muscle organ development; GO: 1901741, positive regulation of myoblast fusion; GO: 0048741, skeletal muscle fiber development) and energy metabolism (GO: 0015991, ATP hydrolysis coupled proton transport; GO: 0097009, energy homeostasis; GO: 0098655, cation transmembrane transport) (Table 4; Figure 4A). 

In contrast, the DEGs identified in the pigs exposed to 25 mg/m^3^ ammonia were enriched in the enzyme activities related to ATP metabolism, lipid metabolism [including hydrolase of myristoyl (GO: 0016295), 3-hydroxypalmitol (GO: 0004317), palmitoyl and oleoyl acyl-carrier proteins (GO: 0016296 and GO: 0004320), reductase of enoyl and oxoacyl acyl-carrier proteins (GO: 0004319 and GO: 0004316)] and DNA binding (GO: 0003677). They were mostly involved in the biological process of cell mitosis (Table 4; Figure 4B). The KEGG pathway analysis revealed several overrepresented pathways in pigs exposed to 10 mg/m^3^, including circadian rhythm (ssc04710) and phagosome (ssc04145), while pathways including cell cycle (ssc04110) and fatty acid metabolism (ssc01212) were enriched in pigs exposed 25 mg/m^3^ (Table 4; Figure 4C,D).

### 3.7. Verification of RNA-Seq by qRT-PCR

We chose nine DEGs to validate RNA-Seq results using qRT-PCR (Figure 5). In pigs exposed to 10 mg/m^3^ ammonia, *MyoG* and *IRS1* were decreased while *NR1D2* and *IL1RAP* were increased. In those pigs exposed to 25 mg/m^3^ ammonia, *FASN*, *CCNB3*, *SCD* were increased, whereas *IL18* was reduced and *FOMX1* was unaltered. Compared to the fold changes in the RNA-Seq, high consistency in fold changes has been found in the results generated by qRT-PCR.

## 4. Discussion

Due to its noxious odor, corrosive characteristics and toxicity, the level of atmospheric ammonia is suggested to be under 17.5 mg/m^3^ (25 ppm) in animal facilities for the sake of both workers and animals [19]. Aerial exposure of ammonia below 25 mg/m^3^ did not necessarily result in increased serum ammonia. Although ammonia can enter the body via gas exchange in the lung, the liver is highly capable of detoxifying ammonia. In the periportal hepatocytes, ammonia can be incorporated into urea, representing a high capacity but a low-affinity means; while in the perivenous hepatocytes, it is converted to glutamine, representing a low capacity but a high-affinity pathway. A properly functional liver can prevent aerial ammonia from accumulating in the body. Despite the fact that serum glutamine and glutamate were not affected by aerial ammonia exposure, glutamine content was reduced in the *Longissimus dorsi* muscle. Glutamine can enter into the citrate cycle via reductive carboxylation to synthesize acetyl-CoA and therefore serves as a lipogenic precursor [20]. Hyperammonemia-induced increments in fatty acids and glycerides can be partially reversed by inhibiting glutamine synthase in the astrocytes [21]. During aerial ammonia exposure, the increased fat content in *Longissimus dorsi* muscle may result from the glutamine flux to lipid.

Many studies have noted the adverse effects of higher atmospheric ammonia on the growth performance of pigs [22,23,24]. In the current study, we found feed intake and body weight gain were not reduced by 10 and 25 mg/m^3^, but some meat quality traits, including muscular fat content and meat color, were influenced by high ammonia exposure. Accordingly, the composition of different myofiber in *Longissimus dorsi* muscle was changed by the high level of aerial ammonia with a dose-dependent manner. Exposure to 10 and 25 mg/m^3^ ammonia results in a transition from oxidative to glycolytic muscle types, indicated by the decreased proportion of *MyHC I* and *IIa* and increased *MyHC IIx* and *IIb*. The oxidative muscle displays a darker color than the glycolytic muscle [25].

Previous studies have demonstrated that hyperammonemia due to malfunctions of the liver, up-regulates *MSTN* expression in the skeletal muscle [7,15]. As a protein synthesis inhibitor, *MSTN* can inhibit the activation of mTOR or downstream molecules either via AKT dependent or independent mechanism, leading to the loss of skeletal muscle mass and the increase of autophagy [5,6]. *MSTN* can also regulate the expression of *FoxO1* and *Atrogin-1* via acting on the downstream molecules of TGF-β/Smad3 signal pathway [16], causing the transition from fast to slow myofibers [26,27,28]. In contrast, aerial ammonia exposure under the level of 25 mg/m^3^ did not increase serum ammonia as well as muscular *MSTN*. The expression of molecules on the action pathway of mTOR was not altered either. Therefore, *MSTN* appears not the key factor for regulating muscle physiology during aerial ammonia exposure.

Using the RNA-Seq, genes involved in muscle development were found to be remarkably affected by aerial ammonia exposure. Among them, myogenic regulatory factors, *MyoD1* and *MyoG*, were down-regulated in pigs exposed to 10 mg/m^3^ ammonia and *MyoD1* was down-regulated in pigs exposed to 25 mg/m^3^ ammonia. *MyoD1* plays a fatal role in the myofiber differentiation by arresting the cell cycle [29]. *MyoD1* can bind to the E-box in the promoter to activate *MyHC IIb* gene expression [30,31]. The expression of *MyHC IIb* is closely related to *MyoD1* [32]. *MyoG* is mainly expressed at the end of differentiation. Manipulating *MyoG* gene expression can affect the differentiation of C2C12 mouse fibroblast cell line [33]. At a low dose, *MyoG* promotes the formation of fast type of myofiber; whereas at a high dose it promotes the production of slow type of myofiber [34]. Because *MyoD* was highly expressed in the fast type of myofibers while *MyoG* was highly expressed in the slow type [35,36], the ratio of these two myogenic factors plays a critical role in the transition of muscle fibers [37,38]. The decreased *MyoD/MyoG* may promote the conversion from slow myofiber to fast myofiber during ammonia exposure.

One of the key characters of meat quality which have been affected by aerial ammonia exposure is fat content in muscle. Previously fat distribution in broiler chickens was noted to be altered by aerial ammonia [11]. Genes participating in steroid biosynthesis and peroxisome proliferator-activated receptor signaling pathway was activated during ammonia exposure, resulting in decreased fat deposition in the breast muscle [10]. In the pigs exposed to 25 mg/m^3^ ammonia, many genes critical to fatty acid synthesis and oxidation were changed when muscle fat was increased. Among them, enzymes catalyzing fatty acid synthesis (including *FASN*, *SCD* and *FADS1*) were up-regulated, while *ACADL*, an enzyme catalyzing β-oxidation, was down-regulated. Fatty Acid Synthase (*FASN*) plays a key role in the *de novo* synthesis of long-chain fatty acids. In mice [39] and chickens [40], liver fat is reduced and fatty acid oxidation is increased when *FASN* gene expression is inhibited [39,41]. Stearoyl-CoA desaturase (*SCD*) catalyzes the last step of the monounsaturated fatty acid synthesis. Disruption of *SCD* gene profoundly reduces the development of obesity by impairing lipid synthesis [42,43]. *FADS1* is a key gene for the synthesis of long-chain unsaturated fatty acids [44]. The level of *FADS1* expression is positively related to fatty acids in the serum and adipose tissue [45,46]. The concomitant up-regulation of *FASN*, *SCD* and *FADS1* genes and down-regulated *ACADL* increased fatty acid synthesis and reduced fatty acids β-oxidation, consequently resulting in increased fat content in *Longissimus dorsi* muscle when pigs were exposed to 25 mg/m^3^ ammonia.

## 5. Conclusions

Collectively, atmospheric ammonia exposure impacted meat quality by increasing fat content, lightening meat color and regulating the skeletal muscle fibers switching. In contrast to hyperammonemia induced by cirrhosis, *MSTN*, the powerful regulator in the muscle physiology, appears not the key factor for regulating muscle metabolism during aerial ammonia exposure. At the level of 10 or 25 mg/m^3^, aerial ammonia promoted the transition from slow oxidative myofiber to fast glycolytic myofiber by altering the expression of myogenic factors. At the level of 25 mg/m^3^, aerial ammonia increased lipid content in the skeletal muscle by up-regulating genes encoding key enzymes for fatty acids synthesis and down-regulating enzymes catalyzing β-oxidation.

## Figures and Tables

**Figure 1 animals-10-00002-f001:**
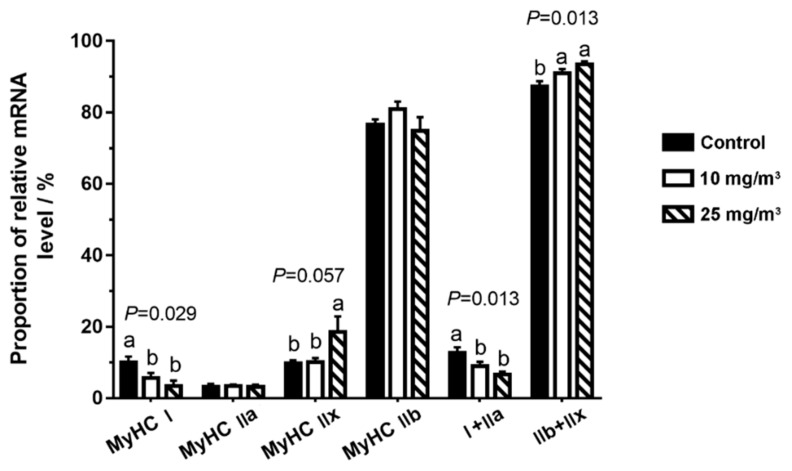
Muscle fiber compositions in the *Longissimus dorsi* muscle of pigs exposed to different aerial ammonia. Data are expressed as means ± SE (n = 6 per group). Bars with different letters are significantly different at *p* < 0.05.

**Figure 2 animals-10-00002-f002:**
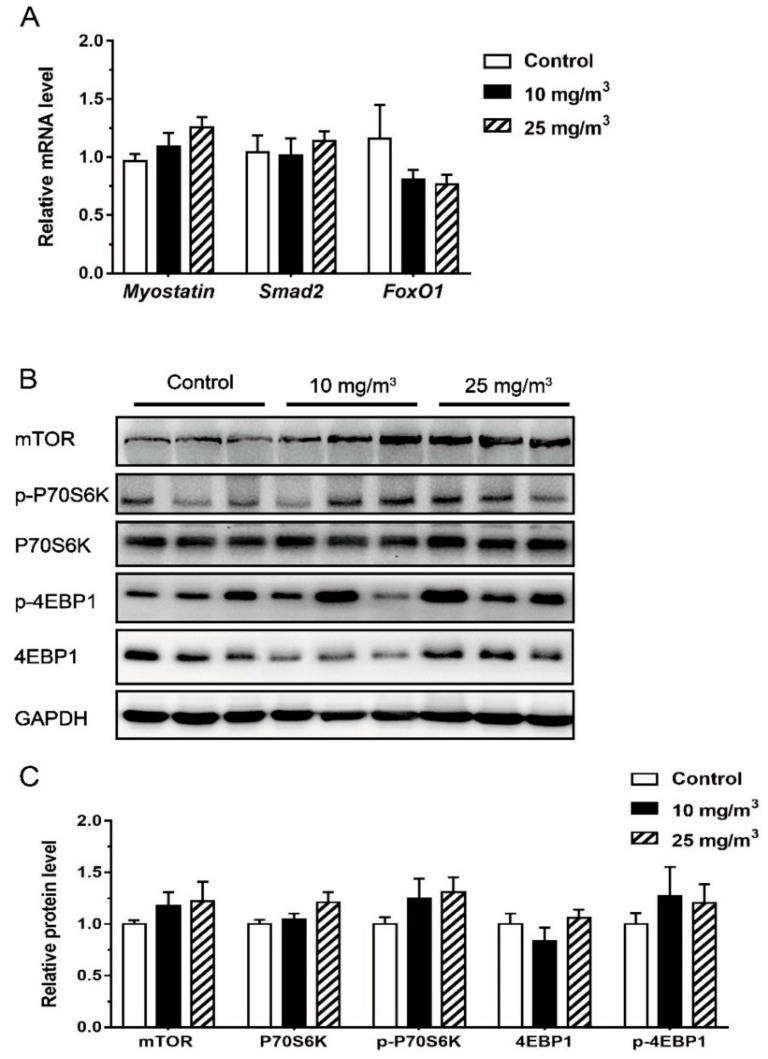
Expression of *MSTN, Smad2, FoxO1* (**A**) and mTOR signal pathway (n = 6 per group) (**B**,**C**) in the *Longissimus dorsi* muscle of pigs exposed to different aerial ammonia. Data are expressed as means ± SE.

**Figure 3 animals-10-00002-f003:**
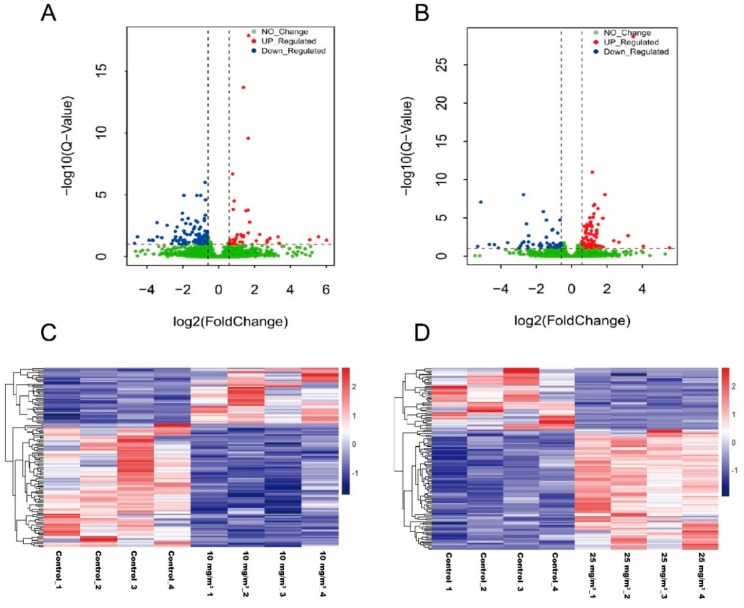
Volcano plots, heat maps and Venn diagram to show differentially expressed genes (DEGs) in *Longissimus dorsi* muscle of pigs exposed to 10 or 25 mg/m^3^ ammonia (n = 4 per group). Volcano plot for control group vs 10 mg/m^3^ group (**A**) and control group vs 25 mg/m^3^ group (**B**); the relative expression level (in FPKM) of differentially expressed genes between control group and 10 mg/m^3^ group (**C**) or between control group and 25 mg/m^3^ group (**D**); Venn diagram between10 mg/m^3^ group vs control group and 25 mg/m^3^ group vs control group (**E**).

**Figure 4 animals-10-00002-f004:**
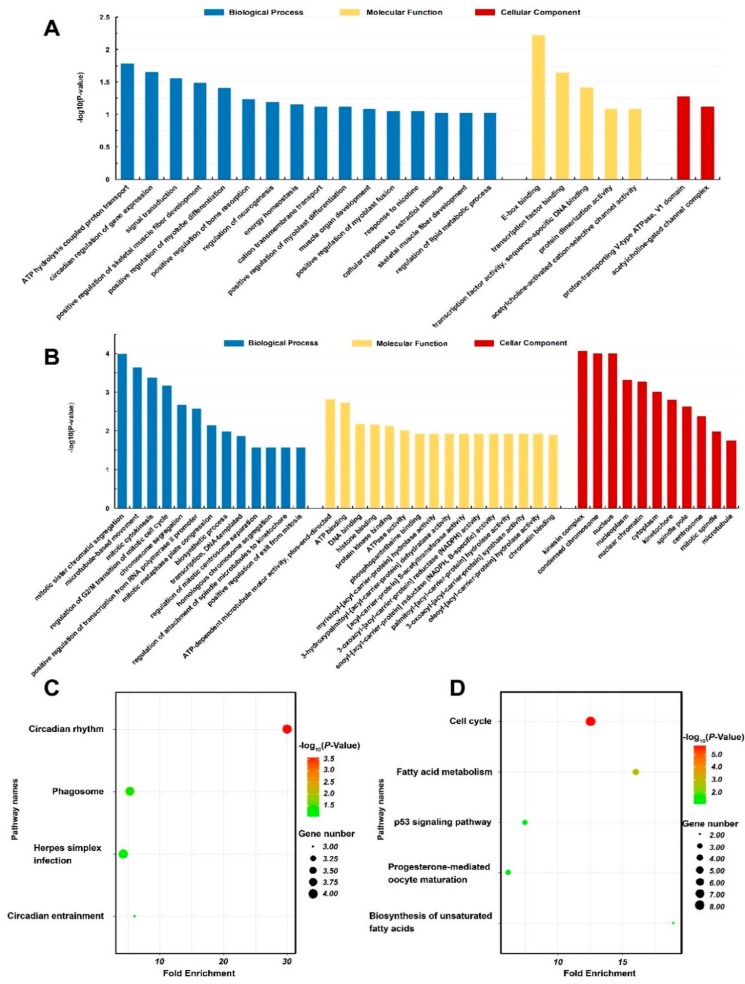
Gene ontology (GO) annotation and KEGG pathway analysis (n = 4 per group). GO annotation of differentially expressed genes in 10 mg/m^3^ group (**A**) or 25 mg/m^3^ group (**B**) compared with control group; KEGG Pathway analysis of differentially expressed genes in 10 mg/m^3^ group (**C**) or 25 mg/m^3^ group (**D**) compared with control group.

**Figure 5 animals-10-00002-f005:**
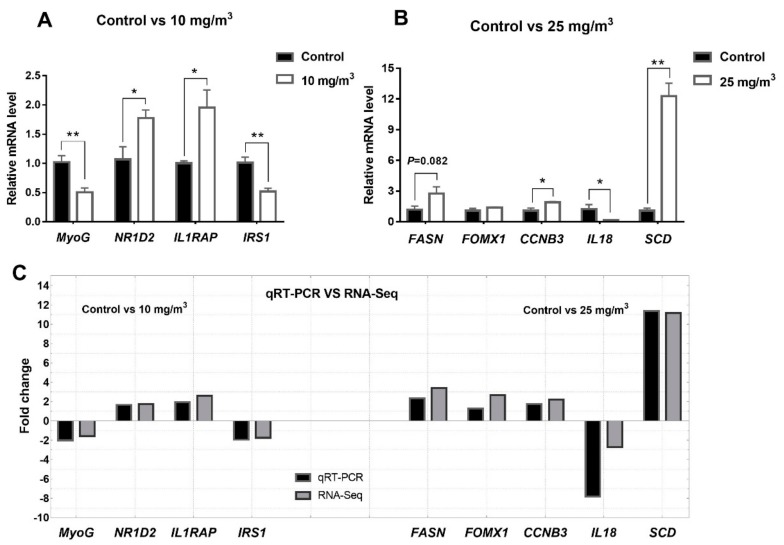
qRT-PCR validation of differentially expressed genes from *Longissimus dorsi* of pigs (n = 4 per group). Differentially expressed genes in control group vs 10 mg/m^3^ group (**A**) and control group vs 25 mg/m^3^ group (**B**); fold change of differential expressed genes between qRT-PCR validation and RNA-Seq analysis (**C**). * Indicates *p* < 0.05 and ** indicates *p* < 0.01, *GAPDH* was used to an internal control.

**Table 1 animals-10-00002-t001:** Growth performance and meat quality of pigs exposed to atmospheric ammonia.

Items	Ammonia Concentration, mg/m^3^			SEM	*p* Value
	0	10	25		
Initial body weight, kg	20.33	21.20	20.72	0.565	0.566
End body weight, kg	30.55	32.70	31.87	1.003	0.338
Body weight gain, kg	10.22	11.25	11.10	0.441	0.478
Average daily feed intake, g/d	1028.02	1104.04	1048.24	34.980	0.310
pH_45 min_	7.01	7.13	7.00	0.061	0.283
Drip loss, %	3.84	3.87	3.37	0.276	0.384
Meat color	85.36 ^a^	87.54 ^a^	77.82 ^b^	1.444	0.001
Fat content, g/100 g freeze-dried muscle	5.81 ^b^	9.05 ^a^	8.84 ^a^	0.683	0.007

^a^^,b^ Values with a row not sharing a common superscript letter indicate a significant difference between groups at *p* < 0.05 (n = 6 per group).

**Table 2 animals-10-00002-t002:** Profile of ammonia, urea and free amino acids in serum (n = 6 per group).

Items	Ammonia Concentration, mg/m^3^			SEM	*p* Value
	0	10	25		
Serum ammonia and urea					
Serum ammonia, umol/L	135.70	135.50	112.97	7.085	0.067
Serum urea, mmol/L	2.94	3.09	3.15	0.167	0.660
Serum-free amino acid, nmol/mL					
Glutamic acid	240.93	234.74	177.90	23.611	0.157
Glutamine	512.90	470.88	517.98	29.658	0.527
Alanine	490.18	484.06	479.92	38.717	0.982
b-Alanine	111.35	107.82	104.08	5.833	0.685
Citrulline	65.17	62.22	64.43	4.840	0.915
Arginine	153.23	195.36	152.37	32.177	0.609
Ornithine	105.22	101.02	101.43	10.606	0.956
Isoleucine	82.48	75.88	83.70	3.586	0.325
Leucine	146.57	147.76	159.48	6.494	0.335
Aspartic acid	42.63	43.68	32.45	3.513	0.082
Asparagine	39.57	36.80	44.62	3.134	0.260
Serine	135.53	122.86	114.68	7.156	0.153
Histidine	48.92	51.22	47.13	3.699	0.762
Glycine	1383.90	838.80	822.43	259.720	0.263
Threonine	95.05	107.44	101.87	10.615	0.736
Taurine	80.00 ^b^	112.40 ^a^	76.00 ^b^	9.508	0.045
Tyrosine	59.15 ^b^	57.64 ^b^	77.17 ^a^	4.358	0.013
Tryptophan	42.78	36.08	43.10	2.913	0.230
Methionine	26.42	22.06	28.12	2.756	0.220
Valine	187.08 ^ab^	155.96 ^b^	198.23 ^a^	10.073	0.037
Phenylalanine	67.88	67.34	73.45	2.851	0.290
Lysine	100.20	105.64	105.18	5.919	0.782

^a,b^ Values with a row not sharing a common superscript letter indicate a significant difference between groups at *p* < 0.05 (n = 6 per group).

**Table 3 animals-10-00002-t003:** Free amino acids in muscle (n = 6 per group).

Items	Ammonia Concentration, mg/m^3^			SEM	*p* Value
	0	10	25		
Glutamic acid	267.54	258.35	282.51	35.637	0.891
Glutamine	5182.77 ^a^	2828.16 ^b^	2336.01 ^b^	577.525	0.008
Alanine	1281.96	1336.08	1111.67	149.186	0.557
Citrulline	121.24	118.13	105.61	9.348	0.493
Arginine	73.17	72.39	69.76	8.509	0.959
Ornithine	73.52	84.44	67.51	12.788	0.647
Isoleucine	51.56 ^a^	44.03 ^a,b^	34.45 ^b^	3.567	0.019
Leucine	*ND*	*ND*	*ND*		

^a,b^ Values with a row not sharing a common superscript letter indicate a significant difference. *ND*: not detectable; unit: umol/g tissue.

**Table 4 animals-10-00002-t004:** Enriched KEGG pathway and GO term of differentially expressed genes in the *Longissimus dorsi* muscle of pigs (n = 4 per group).

Category	Term	Enriched Genes	*p* Value
0 ppm VS 10 mg/m^3^			
GO term	ATP hydrolysis coupled proton transport	*ATP6V1B2*↑*, ATP1A1*↓*, ATP6V1C2*↓	0.017
GO term	Signal transduction	*CHRNG*↓*, LVRN*↓*, ARHGAP20*↓*, RREB1*↓*, IL1RAP*↑	0.027
GO term	Positive regulation of skeletal muscle fiber development	*MYOD1*↓*, MYOG*↓	0.033
GO term	Positive regulation of myotube differentiation	*MAMSTR*↓*, MYOG*↓	0.039
GO term	Energy homeostasis	*MRAP2*↓*, AMPD3*↓	0.070
GO term	Regulation of lipid metabolic process	*NR1D2*↑*, IRS1*↓	0.095
0 ppm VS 25 mg/m^3^			
GO term	ATPase activity	*ABCA1*↓*, ATAD2*↑*, KIF20A*↑*, KIF22*↑	0.010
KEGG pathway	Cell cycle	*CCNA2*↑*, PTTG1*↑*, MCM3*↑*, CCNB1*↑*, E2F1*↑*, CCNB3*↑*, ESPL1*↑*, MCM2*↑	0.000
KEGG pathway	Fatty acid metabolism	*FADS1*↑*, ACADL*↓*, FASN*↑*, SCD*↑	0.002
KEGG pathway	p53 signaling pathway	*GTSE1*↑*, CCNB1*↑*, CCNB3*↑	0.057
KEGG pathway	Biosynthesis of unsaturated fatty acids	*FADS1*↑*, SCD*↑	0.098

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
