# Peer review of "Atmospheric Ammonia Affects Myofiber Development and Lipid Metabolism in Growing Pig Muscle"

_animals, 2019, doi:10.3390/ani10010002_

Round 1
Reviewer 1 Report
The author mentioned that the 4EBP1 protein show one band in pig. However, I highly doubt about it. Just simply listed some References below. Also, please show the results of p-mTOR.
Yin Y, Yao K, Liu Z, et al. Supplementing L-leucine to a low-protein diet increases tissue protein synthesis in weanling pigs[J]. Amino acids, 2010, 39(5): 1477-1486.Wang C, Chen F, Zhang W, et al. Leucine promotes the growth of fetal pigs by increasing protein synthesis through the mTOR signaling pathway in longissimus dorsi muscle at late gestation[J]. Journal of agricultural and food chemistry, 2018, 66(15): 3840-3849.
Ellederova Z, Kovarova H, Melo‐Sterza F, et al. Suppression of translation during in vitro maturation of pig oocytes despite enhanced formation of cap‐binding protein complex eIF4F and 4E‐BP1 hyperphosphorylation[J]. Molecular Reproduction and Development: Incorporating Gamete Research, 2006, 73(1): 68-76.
Davis T A, Nguyen H V, Suryawan A, et al. Developmental changes in the feeding-induced stimulation of translation initiation in muscle of neonatal pigs[J]. American Journal of Physiology-Endocrinology And Metabolism, 2000, 279(6): E1226-E1234.
Chen Y, Zhu H, McCauley S R, et al. Diminished satellite cell fusion and S6K1 expression in myotubes derived from skeletal muscle of low birth weight neonatal pigs[J]. Physiological reports, 2017, 5(3): e13075.
Qin Q, Xu X, Wang X, et al. Glutamate alleviates intestinal injury, maintains mTOR and suppresses TLR4 and NOD signaling pathways in weanling pigs challenged with lipopolysaccharide[J]. Scientific reports, 2018, 8(1): 15124.
Deng D, Yao K, Chu W, et al. Impaired translation initiation activation and reduced protein synthesis in weaned piglets fed a low-protein diet[J]. The Journal of nutritional biochemistry, 2009, 20(7): 544-552.
Reviewer 2 Report
The author has responded to the comments well. It suggested to be published in Animals.
Reviewer 3 Report
The authors have carefully revised the manuscript, but in order to ensure the quality of the article, the authors should carefully check some details in the article.
Reviewer 4 Report
The paper can be published in its current form
Round 2
Reviewer 1 Report
Please re-run your total 4EBP1 in 15% gel. Please re-run p-mTOR. This is an important data.This manuscript is a resubmission of an earlier submission. The following is a list of the peer review reports and author responses from that submission.
Round 1
Reviewer 1 Report
This manuscript describes the effect of atmospheric ammonia on meat quality and fat metabolism. They show some interesting data in this paper, but some major issues need to be addressed.
Major:
With the increase of the environmental ammonia, the author sees the decrease of the serum ammonia. Why? Muscle staining might need to verify the change of type I or type 2 fiber in muscle. For WB (2C), from mTOR and S6K1 results, I feel the loading of different samples is not equal (usually mTOR and S6K1 should be same). If the author did a lighter exposure film, they might see the difference of GAPDH among different treatments. Furthermore, 4EBP1 have three bands, but not signal one band. you can check this antibody information from CST website or from other published data. For WB (2E), I would expect the error bars should larger when authors quantified this WB. Usually, 2E and 2C should be combined together.
Minor:
‘ppm’ is not suitable for a scientific paper Is table 2 presented in correctly? It seems so confusing.Author Response
Please see the attachment.

Reviewer 2 Report
The paper "Atmosferic ammonia affects meat quality in pigs by regulating myofiber development and lipid metabolism" is very interesting.
In fact is well write and, in my opinion, must be accept after minor revision.
1) Please note corrections to minor methodological errors
2) please carefully check the text editing.
Reviewer 3 Report
The manuscript by Tang et al. investigated the effect of aerial ammonia on meat quality, and found there are different mechanisms at concentration of 15 and 32 ppm of aerial ammonia by altering the expression of myogenic factors, or up-regulating genes encoding key enzymes for fatty acids synthesis and down-regulating enzymes catalyzing β-oxidation. The results are interesting, the paper is well written, and the scope generally fits within the scope of the journal.
Specific comments:
What was the hypothesis of this research? What is the basis for choosing the ammonia concentration 15 and 32 ppm? Table 2: It is nmol/mL, not nmol/ml. Figure 4: The image and label in x-axis is not clear.
Reviewer 4 Report
The authors want to study the mechanism of ammonia on meat quality, which is innovative to some extent. However, there are many problems in the paper, as follows:
The authors have explored the influence of 15ppm and 32ppm of ammonia on pork quality, but why choose these two concentrations? In this paper, when pigs were exposed to 32ppm ammonia, their body weight, average body weight gain, water drop loss and pH45 did not change. Only the decrease of flesh color brightness and mRNA level of myosin heavy chain isoforms , which could not fully show that 32ppm affected pork quality through affecting muscle fiber type. The authors the authors exposed 70 - day - old pigs for 25 d in 0 ppm, 15 ppm and 32 ppm of ammonia to explore the effect of ammonia on intramuscular fat, but The major increase in intramuscular fat began after 16 weeks (112 d) of age (Lee B, Kauffman R. G. Cellular and Enzymatic Changes with Animal Growth in Porcine Intramuscular Adipose Tissue. Journal of Animal Science, 1974, Volume 38, Issue 3, March, Pages 532–537). Therefore, I personally think it is inappropriate. The authors expected to study the mechanism of ammonia's negative effect on meat quality. But they only detected the expression of FoxO1, MSTN and the downstream mTOR pathway. and None of them changed significantly. Therefore, the authors considered that the effect of ammonia on meat quality is not via the FoxO1 signaling pathway. But FoxO1 is known to function through phosphorylation. Therefore, I personally think it is inappropriate. To reveal the mechanism of ammonia affecting meat quality, RNA-seq was employed to screen for important genes associated with ammonia exposure. But the authors did not analyze the downstream pathways of significantly altered genes in detail. So the result did not achieve the desired purpose. In addition, there are many small problems in the article, such as, in line 215, the contents of “Table2-------“and “Table3------“are unclear; in lines 241-243, “a total of 123------,which includes 85 down-upregulated and 35 up-regulated-------“, on the one hand, there is a clerical error,” down-upregulated”, on the other hand, the sum of the Numbers is wrong ”85+35=120”; in line 284,”-----CCNB3,SCD were increased, whereas CCNB3 was reduced------“ So, is CCNB3 increased or induced?
